# MALTectomy and psoriasis risk in women: A prospective study in the French E3N prospective cohort

Marco Conte[1], Agnes Fournier[1], Joseph A. Rothwell[1], Marie-Christine Boutron-Ruault[1], Laura Baglietto[2], Marco Fornili[2], Emilie Sbidian[3,4], Gianluca Severi[1,5]*

1 UVSQ, Inserm, Gustave Roussy, Exposome and Heredity Team, Centre for Epidemiology and Population Health (CESP U1018), Université Paris-Saclay, Villejuif, France, 2 Department of Clinical and Experimental Medicine, University of Pisa, Pisa, Italy, 3 Department of Dermatology, Hôpital Henri Mondor, Créteil, France, 4 Epidemiology in Dermatology and Evaluation of Therapeutics (EpiDermE) - EA 7379, Université Paris Est Créteil (UPEC), Créteil, France, 5 Department of Statistics, Computer Science and Applications « G. Parenti », University of Florence, Italy

* Gianluca.severi@inserm.fr

## Abstract

### Background and aim

The involvement of mucosa associated lymphoid tissues (MALT) in the development of an autoimmune response in the skin is unclear and unstudied. In this study we sought to assess the relationship between removal of MALT tissues (MALTectomy) and the risk of development of psoriasis (overall or moderate-to-severe).

### Methods

We conducted a prospective observational study based on E3N, a French cohort composed of 98 995 women born between 1925 and 1950 and insured by the health insurance of the national education system (MGEN). The study population included the 90 119 women that completed the 1990 baseline questionnaire with available information on MALTectomy and a valid incident diagnostic date for psoriasis. During the 1990–2018 follow-up period 2 433 incident cases of psoriasis were identified through self-reports while during the period for which drug reimbursement data were available from the MGEN database (2004–2018), 120 cases of moderate-to-severe psoriasis were identified. Hazard ratios (HR) and their 95% confidence intervals were estimated by Cox proportional hazards models and adjusted for known or putative psoriasis risk factors.

### Results

Appendectomy was associated with an increased risk of psoriasis both in the univariate [HR: 1.17 (95% CI: 1.08–1.27)] and multivariable models [HR: 1.14 (95% CI: 1.05–1.24)]. A suggestive association with appendectomy was observed for moderate-to-severe psoriasis risk [univariate HR: 1.40 (95%CI: 0.97–2.02); multivariable HR: 1.36 (95%CI: 0.94–1.96)]. No association was observed between tonsillectomy or adenoidectomy both for overall and moderate-to-severe psoriasis.

**Data Availability Statement:** Data underlying this article are made available under managed access owing to governance constraints and need to protect the privacy of study participants. Data on

E3N cohort participants are available to bona fide researchers for all types of health-related research, which is in the public interest. Raw data requests should be submitted through the E3N website (www.e3n.fr) or sent to contact@e3n.fr and will be reviewed by the E3N Access Committee. Further information is provided at https://www.e3n.fr/node/78.

**Funding:** The authors received no specific funding for this work.

**Competing interests:** The authors have declared that no competing interests exist.

## Conclusions

The observed association between appendectomy and risk of psoriasis warrants further investigations as they may help to elucidate the disease etiology and improve risk prediction.

## Introduction

Psoriasis is a common chronic inflammatory skin disease characterized by a wide variety of symptoms and severity [1]. It is caused by a network of genetic, immunological, and environmental factors that is not fully understood but where lifestyle plays a role with body mass index (BMI) [2], smoking and diet implicated as risk factors as observed in longitudinal cohort studies [3, 4]. The main pathogenetic mechanism of psoriasis is a lack or reduced activity of peripheral regulatory T cells [5–7] that are responsible for immunological tolerance by suppressing T-helper cells directed against self-antigens of the skin. Maturation of regulatory T cells occurs primarily in the thymus [8], but also in peripheral tissues such as the mucosa-associated lymphoid tissues (MALT) [9] that are lymphoid structures dispersed in the digestive and respiratory tract, with a structure similar to lymph nodes [10].

For a long period, the role of MALT-associated organs, such as the adenoid tonsils, palatine tonsils, and appendix, was considered to be limited to a local immune response [11].

In the last 20 years, many studies have shed light on the role of MALT tissue in the development of the systemic immune response: the adenoids, tonsils, and appendix all harbor extra-thymic T cells expressing the autoimmune regulator gene (Aire) (eTACs) [12] involved in T cell selection and tuning of peripheral tolerance [13, 14]. Furthermore, MALT could play a pivotal role in microbiota homeostasis [15–17].

Removal of MALT tissue, or MALTectomy, such as adenoidectomy, appendectomy, and tonsillectomy may affect the composition of the microbiota, [18] and dysregulate peripheral tolerance, which constitutes the basis of autoimmunity [19].

Despite the important role of MALT tissue on systemic immunological homeostasis, it is unknown whether MALTectomy is risk factors of autoimmunity.

While experimental studies have permitted significant progress in the treatment of autoimmune diseases, epidemiological studies are still critical to fully elucidate the role of risk factors in the pathogenesis of these disorders.

For this reason, we explored the potential associations between MALTectomy and skin autoimmunity and specifically on the risk of psoriasis, overall or moderate-to-severe, using the French E3N prospective cohort that has been extensively used to study risk factors for autoimmune diseases [20, 21].

## Materials and methods

This is a prospective observational cohort study with different types of MALTectomy (appendectomy, adenoidectomy and tonsillectomy) as the exposure of interest and incident psoriasis as the outcome. This investigation was based on data from the E3N cohort.

### E3N cohort

E3N (Étude Épidémiologique auprès de femmes de l'Éducation Nationale) is a French prospective cohort. It was created in 1990 through the inclusion of women born between 1925

and 1950 and affiliated to the Mutuelle Générale de l'Éducation Nationale (MGEN), the French health insurance scheme covering workers in the national education system, mostly teachers. Participants completed the baseline and follow-up questionnaires sent by mail every two to three years including questions on medical history, reproductive life and lifestyle (e.g., nutrition and physical activity).

Further details about the cohort are available elsewhere [22]. Response rates were around 80–85% for each follow-up questionnaire. All participants in the cohort provided an informed consent.

## Identification of psoriasis cases

Psoriasis cases within the E3N cohort were ascertained using self-reported information from two main sources: follow-up questionnaires and MGEN drug reimbursements database. First, in follow-up questionnaires sent in 2008, 2011, and 2018, participants were asked whether they have ever been diagnosed with psoriasis and when. We resolved inconsistencies in age at onset of the disease declared in different questionnaires, by retaining the age at onset declared in the earliest questionnaire. Moderate-to-severe cases of psoriasis were identified using information from the MGEN drug reimbursement database linked to the cohort that includes vital status, date of death, as well as data on all extra-hospital drug reimbursements (code of the drugs and corresponding dates of purchase) from 2004 onwards for the entire cohort population. Five classes of drugs are used to treat psoriasis (topical drugs derived from vitamin D, topical steroids, psoralens used for phototherapy, non-biologic immunosuppressants, and biologic immunosuppressants; data in S1 File).

Moderate-to-severe cases of psoriasis was defined as self-reported psoriasis with a reimbursement (at any time) for at least one of the above-mentioned systemic drugs (i.e., psoralens, non-biologic immunosuppressants, and biologic immunosuppressants).

## Assessment of MALTectomy

Data on appendectomy, adenoidectomy and tonsillectomy were collected with the baseline questionnaire in 1990 and, only for appendectomy, with the 1992 and 1993 questionnaires. At baseline, participants were asked to declare whether or not they had undergone such surgical operations and the age at the time of the surgical procedure from a list of 6-time intervals (age at MALTectomy -Adenoidectomy, appendectomy, tonsillectomy: ≤10 years; 11–14; 15–19; 20–29; 30–39; ≥40).

## Assessment of covariates

The choice of covariates was based on the availability of data from the E3N cohort on known and putative risk factors for psoriasis that may confound the association between MALTectomy and psoriasis risk. They were chosen based on factors found to be associated with psoriasis in previous studies and on sociodemographic, hormonal and reproductive factors that may be associated with both MALTectomy and psoriasis.

Education and marital status were only obtained from the baseline questionnaire while data on tobacco smoking and BMI were collected at multiple time points from the baseline and all follow-up questionnaires. Height, expressed in centimeters, was collected at baseline (1990), in 1995 and then in every questionnaire returned between 2000 and 2018, while self-reported weight, expressed in kilograms, was collected in each questionnaire. Data on a selection of diseases that may be associated with psoriasis risk (i.e.: depression or anxiety requiring a pharmacological treatment, hypertension and type-2 diabetes) were collected at multiple follow-up questionnaires. Information on menopausal status updated at each questionnaire was used to

define menopausal status and age at menopause. Other reproductive and hormonal factors such as age of menarche, parity, and ever use of the contraceptive pill were collected at the baseline questionnaire.

## Study population

From the initial cohort of 98 995 participants, we excluded all participants with missing values for appendectomy, tonsillectomy, or adenoidectomy (n = 2 493), all participants who died during follow-up but with no valid date of death provided by the MGEN (n = 44) and participants who declared a date of psoriasis diagnosis after 2004 but for whom no reimbursement for drugs used in psoriasis was found in the MGEN database (N = 124). Additional exclusion criteria were applied based on whether the outcome of interest was overall or moderate-to-severe psoriasis. Overall psoriasis analyses included 90 119 participants after the exclusion: i) participants who did not answer any other questionnaire after baseline (n = 2 813), ii) all prevalent cases of psoriasis at baseline (n = 1 276), iii) those with missing age at diagnosis (n = 1 137), iv) psoriasis cases identified only through reimbursement for topical drugs derived from vitamin D in the MGEN drug database, for whom it was not possible to determine a date of diagnosis (n = 989) (see Fig 1).

The analyses on the risk of developing moderate-to-severe psoriasis included 78 269 participants after the exclusion of participants that were dead or lost to follow up by January 1$^{st}$, 2004, when drug reimbursement data became available for the E3N database (n = 14 121), the exclusion of psoriasis cases occurred before January 1$^{st}$, 2004 (n = 2 116) and the exclusion of psoriasis cases for which it was not possible determine diagnostic date (N = 1 828) (see Fig 2). The comparison between women included and excluded from the overall psoriasis analysis population is reported in S1 File. Concisely, excluded women did not materially differ from included women regarding MALTectomy history. However, excluded women were less educated, had a higher BMI and were more often current smokers at baseline.

## Statistical analysis

To test for associations between psoriasis risk and the different types of MALTectomy, hazard ratio (HR) and 95% confidence interval (CI) were estimated using Cox proportional hazard regression models.

Participant age was used as the time-scale. Follow-up time for the study started at the date when participants returned the baseline questionnaire sent in 1990, and ended at the earliest date at psoriasis diagnosis, at death, at the last answered questionnaire, or at the date of the questionnaire mailed in 2018 (last E3N questionnaire used for this study) was returned, whichever occurred first.

For the analyses on moderate-to-severe psoriasis, follow-up started on January 1$^{st}$, 2004, and ended at the earliest date when a moderate-to-severe psoriasis status occurred, at death, or on December 31$^{st}$, 2018 (date of the most recent available data on drug reimbursement), whichever occurred first. Each type of MALTectomy (adenoidectomy, appendectomy, and tonsillectomy) was modelled as a fixed binary variable taken from baseline A graphical representation of the cohort follow-up and of the data sources is presented in Fig 3.

We fitted both univariate models and multivariable models adjusted for potential risk factors for psoriasis [23] or for socio-economic variables that could act as confounders.

Univariate models were age-adjusted while multivariable models included BMI (categorical variable: ≤24, 25–29, and ≥ 30 kg/m2), marital status (unmarried, married), smoking status (never, current or former smoker), level of education (undergraduate or less, graduate, postgraduate or more), and three binary variables about medical conditions: diabetes, depression

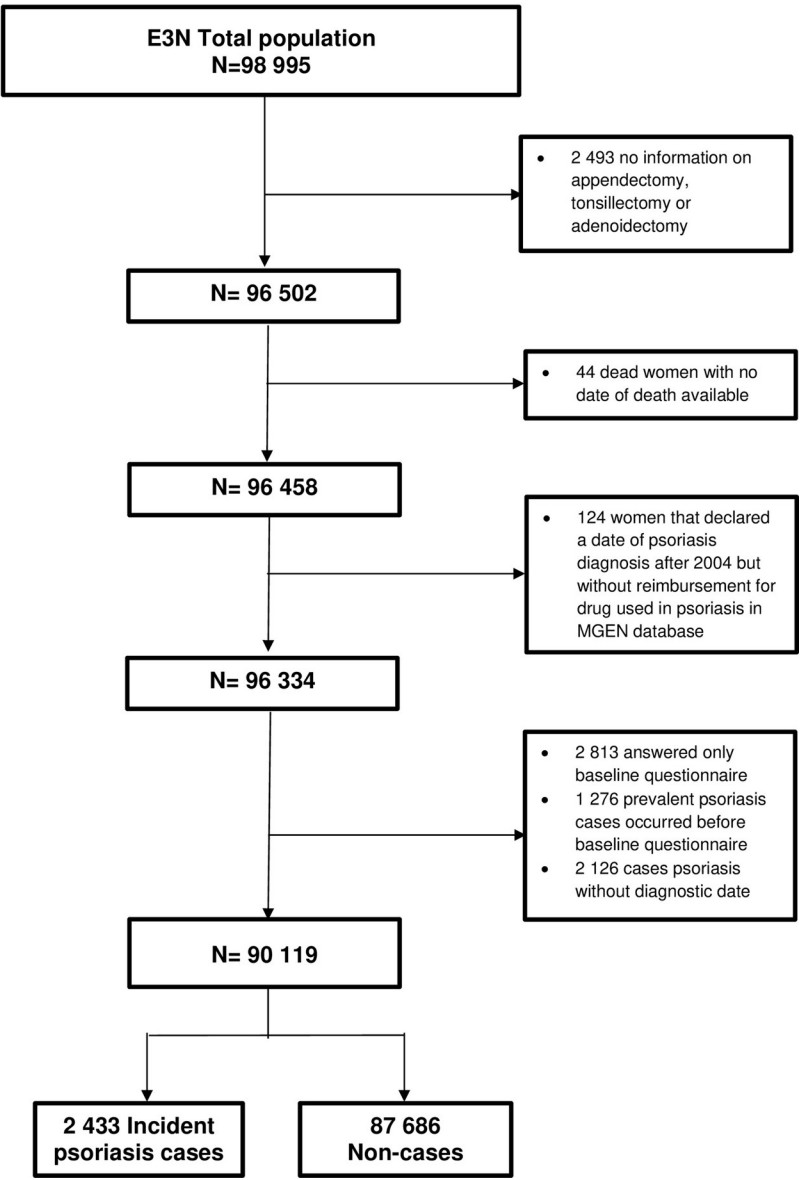

**Fig 1. Flow chart of the included participants in overall psoriasis models.**

or anxiety, and hypertension. Both univariate and multivariable models were fitted including only one type of MALTectomy at a time and the absence of the specific type of MALTectomy was used as the reference category in all models. Potential effects of reproductive and hormonal factors collected in E3N on the association between MALTectomy and psoriasis were explored including age at menarche (≤11 years, 12–14 years, ≥15 years), menopausal status (premenopausal menopausal), parity (nulliparous, parous), and ever use of a contraceptive pill (no, yes).

BMI, smoking status, menopausal status, diabetes, depression, and hypertension were modelled as time-varying variables.

Additional models were fitted including age at MALTectomy. We modelled appendectomy as a time-varying variable in the multivariable model to account for cases occurred after

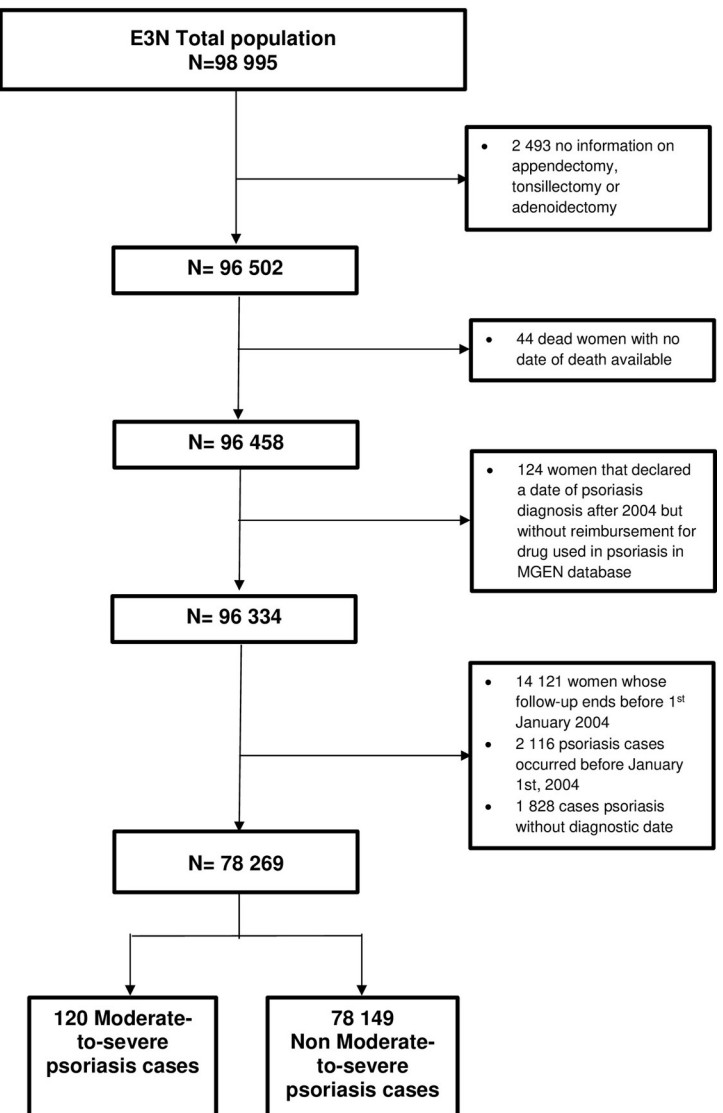

**Fig 2. Flow chart of the included participants in moderate-to-severe psoriasis models.**

baseline. A further model including alcohol consumption was fitted using 70 350 participants for whom data on alcohol consumption was available from the 1993 questionnaire.

The proportional hazards assumption was verified through the analysis of Schoenfeld residuals. In case of missing values for a time varying variable we used the last observation carried forward (LOCF) method where possible. In case of less than 5% of missing values for a variable, we imputed the mean or the median for quantitative and qualitative variables, respectively. Finally, to evaluate whether exclusions from the study population from the principal models may have affected the results, we performed a sensitivity analysis by using all cases irrespective of whether they were prevalent or incident and including cases with unknown age of diagnosis. For this analysis we used a population of 96 458 and a total of 6 030 psoriasis cases and we fitted univariate and multivariable logistic regression to estimate odds ratio (OR) and 95% confidence interval (CI).

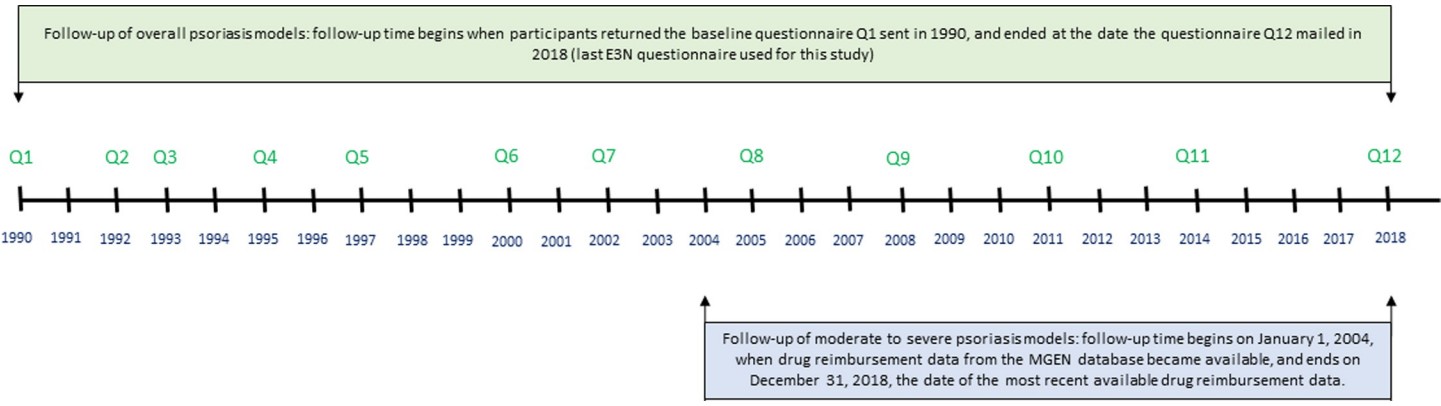

**Fig 3. Graphical representation of the cohort follow-up.**

All analyses were performed between May and August 2021 using SAS version 9.4 (SAS Institute Inc).

Study approval was obtained from the French National Commission for Data Protection and Individual Freedom (327346-V14) and the French Advisory Committee on Information Processing in Material Research in the Field of Health (13.794). All participants signed a written informed consent form at the study entry. Authors had no access to information that could identify individual participants during or after data collection.

## Results

During a median follow-up of 23.8 years (2 145 449 person-years), 2 433 incident cases of psoriasis were recorded, or an annual incidence rate of 113 cases per 100 000. Of these cases, 2 201 (90.4%) had reimbursements for medications consistent with anti-psoriatic treatment. Among these 1 144 (47.0%) had at least one prescription of topical drugs derived from vitamin D, 2 153 (88.5%) for topical steroids, 59 (2.4%) for psoralens, 163 (6.6%) for non-biologic immunosuppressants, and 23 (0.9%) for biologic immunosuppressants. Participants were often treated with combinations of the above-mentioned drugs. The mean (SD) time between baseline and psoriasis diagnosis was 16.5 (±5.9) years, and the mean age at psoriasis diagnosis was 49.9 (±6.6) years.

The characteristics of the study population are reported in Table 1 by psoriasis status and in in S1 File by MALT status.

Cases have a slightly greater BMI and more likely to have a history of smoking than non-cases.

In the population for moderate-to-severe psoriasis analyses, of the 6 030 cases of psoriasis identified in the cohort before and after baseline, 120 participants were classified as moderate-to-severe based on their drug prescription history. Data are reported in S1 File.

### Association between MALTectomy and psoriasis

We found a moderate association with risk of developing psoriasis in all models for appendectomy [multivariate model HR: 1.14 (95%CI: 1.05–1.24)] but not for adenoidectomy and tonsillectomy (Table 2).

We found a suggestive association with risk of moderate-to-severe psoriasis for appendectomy [univariate model HR: 1.40 (95%CI: 0.97–2.02)] (Table 3).

**Table 1. Characteristics of study participants according to overall psoriasis status at the end of follow-up (n = 90 119), E3N cohort, France, 1990–2018.** Data are reported as mean (standard deviation) number (percentages) of participants.

| | Psoriasis status[a] | | No psoriasis (n = 87 686) |
|---|---|---|---|
| | All (n = 90 119) | Psoriasis (n = 2 433) | |
| **Age at baseline** | 49.4 (±6.6) | 48.5 (±6.0) | 49.4 (±6.6) |
| **Appendectomy** | | | |
| No | 55731 (61.8) | 1 417 (58.2) | 54 314 (61.9) |
| Yes | 34388 (38.2) | 1 016 (41.7) | 33 372 (38.1) |
| **Adenoidectomy** | | | |
| No | 65315 (72.5) | 1 730 (71.1) | 63 585 (72.5) |
| Yes | 24804 (27.5) | 703 (28.9) | 24 101 (27.5) |
| **Tonsillectomy** | | | |
| No | 61720 (68.5) | 1 653 (67.9) | 60 067 (68.5) |
| Yes | 28399 (31.5) | 780 (32.1) | 27 619 (31.5) |
| **Year of birth** | | | |
| >1945 | 30359 (33.7) | 887 (36.4) | 29 472 (33.6) |
| 1940–1945 | 21780 (24.2) | 659 (27.1) | 21 121 (24.1) |
| 1935–1940 | 17768 (19.7) | 482 (19.9) | 17 286 (19.7) |
| 1930–1935 | 12060 (13.4) | 270 (11.1) | 11 790 (13.4) |
| <1930 | 8152 (9.0) | 135 (5.5) | 8 017 (9.1) |
| **BMI at baseline** | | | |
| BMI<25 | 73861 (82.0) | 1 944 (79.9) | 71 917 (82.0) |
| 25<BMI<30 | 13417 (14.9) | 385 (15.8) | 13 032 (14.9) |
| BMI>30 | 2841 (3.2) | 104 (4.3) | 2 737 (3.1) |
| **Smoking status at baseline** | | | |
| Never | 48735 (54.1) | 1 187 (48.7) | 47 551 (54.2) |
| Current | 13383 (14.9) | 451 (18.6) | 12 929 (14.7) |
| Former | 28001 (31.1) | 795 (32.7) | 27 206 (31.1) |
| **Marital status at baseline** | | | |
| Unmarried | 15917 (17.7) | 456 (18.7) | 15 461 (17.6) |
| Married | 74202 (82.3) | 1 977 (81.3) | 72 225 (82.4) |
| **Education level** | | | |
| Undergraduate or less | 3097 (3.4) | 79 (3.2) | 3 018 (3.5) |
| Graduate | 12056 (13.4) | 276 (11.3) | 11 780 (13.4) |
| Postgraduate or more | 74966 (83.2) | 2 078 (85.5) | 72 888 (83.1) |
| **Age at menarche** | | | |
| 12–15 years old | 68244 (75.7) | 1 788 (73.4) | 66 456 (75.8) |
| <12 years old | 18337 (20.3) | 555 (22.8) | 17 782 (20.3) |
| >15 years old | 3538 (4.0) | 90 (3.8) | 3 448 (3.9) |
| **Menopause status at baseline** | | | |
| Premenopausal | 46142 (51.2) | 1 369 (56.2) | 44 773 (51.1) |
| Postmenopausal | 43977 (48.8) | 1 064 (43.8) | 42 913 (48.9) |
| **Nulliparous status** | | | |
| No | 79568 (88.3) | 2 116 (86.9) | 77 452 (88.3) |
| Yes | 10551 (11.7) | 317 (13.1) | 10 234 (11.7) |
| **Ever use of oral contraceptives at baseline** | | | |
| No | 40450 (44.9) | 1 040 (42.7) | 39 410 (44.9) |
| Yes | 49669 (55.1) | 1 393 (57.3) | 48 276 (55.1) |
| **Pharmacological classes of drugs prescribed from MGEN database** | | | |

*(Continued)*

**Table 1.** (Continued)

| | Psoriasis status[a] | | |
| | All (n = 90 119) | Psoriasis (n = 2 433) | No psoriasis (n = 87 686) |
|---|---|---|---|
| Topical drugs derived from vitamin D | 2 732 (3.0) | 1 144(47.0) | 1 588 (1.8) |
| Biologic immunosuppressants | 209 (0.2) | 23 (0.9) | 186 (0.2) |
| Non-biologic immunosuppressants | 1 551 (1.7) | 163 (6.6) | 1 388 (1.5) |
| Psoralens | 165 (0.2) | 59 (2.4) | 106 (0.1) |
| Topical steroids | 57 817 (64.2) | 2 153 (88.5) | 55 664 (63.4) |

Adenoidectomy and tonsillectomy were not associated with risk of moderate-to-severe psoriasis. Estimated HRs from the multivariable models were virtually the same as those from the univariate models.

Logistic models show an association between appendectomy and psoriasis in both the univariate [OR: 1.08 (95%CI: 1.02–1.14)] and multivariable models [OR: 1.07 (95%CI: 1.01–1.13)]. No association were observed between tonsillectomy or adenoidectomy and risk of psoriasis (Table 4).

Further adjustment for alcohol consumption, the inclusion of age at MALTectomy or the inclusion of appendectomy as time-varying variable did not materially change the associations between MALTectomy and psoriasis risk. Results are shown in tables in S1 File.

## Discussion

Our study investigated the relationship between MALTectomy and incidence of psoriasis in a large prospective cohort of women. We observed a moderate increase in risk of psoriasis associated with appendectomy in the overall analyses while analyses limited to moderate-to-severe psoriasis suggest an increased risk although confidence intervals include the possibility of no effect. The association was nominally slightly stronger for moderate-to-severe psoriasis than for overall psoriasis.

We did not find any association between adenoidectomy, tonsillectomy and risk of psoriasis.

### Comparison with previous studies

Only a few studies have investigated the association between MALTectomy and autoimmune diseases. Tonsillectomy has been extensively described as treatment in patients with both tonsillitis and psoriasis leading to a reduction in psoriasis manifestations [24], but it is still not clear whether tonsillectomy reduces risk of developing psoriasis: one study reported a modest positive association (SIR-Standardized incidence ratio = 1.19, 95% CI 1.08–1.31) [25], while a recent study from Taiwan reported an inverse association between tonsillectomy and psoriasis (HR = 0.43, 95% CI 0.22–0.87) [26]. Studies have been conducted to investigate the association between appendectomy and the risk of other conditions but none on psoriasis. A positive association has been reported between appendectomy and risk of Parkinson disease [27] and multiple sclerosis. For the latter, a meta-analysis on 33 case-control studies reported an increased risk (OR-Odds ratio = 1.16, 95% CI 1.01–1.34) in both women and men who underwent appendectomy before age 20 [28]. Conversely, an inverse association with appendectomy has been observed for ulcerative colitis [29] and Crohn's disease [30], while no association was observed for rheumatoid arthritis [31, 32].

**Table 2. Hazard ratios of psoriasis risk according to MALTectomy history, (n = 90 119), E3N cohort, France, 1990–2018.**

| Exposure | No psoriasis cases (N = 87 686) | Psoriasis cases (N = 2 433) | Hazard Ratio (95% CI) | | | |
| --- | --- | --- | --- | --- | --- | --- |
| | | | Univariate model[a] | p-value | Multivariable model[b] | p-value |
| Appendectomy | | | | | | |
| No | 54 314 | 1417 | 1.00 [Reference] | | 1.00 [Reference] | |
| Yes | 33 372 | 1016 | **1.17 (1.08–1.27)** | <0.001 | **1.14 (1.05–1.24)** | <0.001 |
| Adenoidectomy | | | | | | |
| No | 63 585 | 1 730 | 1.00 [Reference] | | 1.00 [Reference] | |
| Yes | 24 101 | 703 | 1.06 (0.97–1.16) | 0.15 | 1.04 (0.95–1.13) | 0.33 |
| Tonsillectomy | | | | | | |
| No | 60 067 | 1 653 | 1.00 [Reference] | | 1.00 [Reference] | |
| Yes | 27 619 | 780 | 1.02 (0.94–1.11) | 0.29 | 0.99 (0.91–1.08) | 0.92 |

a) Univariate model was age adjusted

b) Multivariate model was adjusted for BMI, smoking status, education level, marital status at baseline, hypertension, depression, diabetes, age at menarche, menopause status, nulliparous status, ever use of contraceptive pill.

Abbreviations: MALT-Mucosa associated lymphoid tissues

**Table 3. Hazard Ratios of moderate-to-severe psoriasis risk according to MALTectomy history, (n = 78 269), E3N cohort, France, 2004–2018.**

| Exposure | Non- cases of moderate-to-severe (N = 78 149) | Moderate-to-severe psoriasis cases (N = 120) | Hazard Ratio (95% CI) | | | |
| --- | --- | --- | --- | --- | --- | --- |
| | | | Univariate model[a] | p-value | Multivariable model[b] | p-value |
| Appendectomy | | | | | | |
| No | 48 445 | 64 | 1.00 [Reference] | | 1.00 [Reference] | |
| Yes | 29 704 | 56 | 1.40 (0.97–2.02) | 0.06 | 1.36 (0.94–1.96) | 0.10 |
| Adenoidectomy | | | | | | |
| No | 56 614 | 84 | 1.00 [Reference] | | 1.00 [Reference] | |
| Yes | 21 535 | 36 | 1.07 (0.72–1.61) | 0.71 | 1.04 (0.70–1.56) | 0.82 |
| Tonsillectomy | | | | | | |
| No | 53 547 | 83 | 1.00 [Reference] | | 1.00 [Reference] | |
| Yes | 24 602 | 37 | 0.89 (0.59–1.33) | 0.58 | 0.86 (0.57–1.29) | 0.47 |

a) Univariate model was age adjusted

b) Multivariate model was adjusted for BMI, smoking status, education level, marital status at baseline, hypertension, depression, diabetes, age at menarche, menopause status, nulliparous status, ever use of contraceptive pill.

Abbreviations: MALT-Mucosa associated lymphoid tissues

**Table 4.  Logistic analyses of overall psoriasis risk according to MALTectomy history, (n = 96 458), E3N cohort, France, 2004–2018.**

| Exposure | Non- cases of psoriasis (N = 90 428) | Psoriasis cases (N = 6 030) | Odds Ratio (95% CI) | | | |
|---|---|---|---|---|---|---|
| | | | Univariate model[a] | p-value | Multivariable model[b] | p-value |
| Appendectomy | | | | | | |
| No | 55 984 | 3 615 | 1.00 [Reference] | | 1.00 [Reference] | |
| Yes | 34 444 | 2 415 | 1.08 (1.02–1.14) | <0.01 | 1.07 (1.01–1.13) | 0.01 |
| Adenoidectomy | | | | | | |
| No | 65 601 | 4 343 | 1.00 [Reference] | | 1.00 [Reference] | |
| Yes | 24 827 | 1 687 | 1.01 (0.96–1.08) | 0.51 | 1.00 (0.94–1.06) | 0.91 |
| Tonsillectomy | | | | | | |
| No | 61 938 | 4 117 | 1.00 [Reference] | | 1.00 [Reference] | |
| Yes | 28 490 | 1 913 | 1.00 (0.94–1.06) | 0.93 | 0.98 (0.93–1.04) | 0.68 |

a) Univariate model was age adjusted

b) Multivariable model was adjusted for BMI, smoking status, education level, marital status at baseline, hypertension, depression, diabetes, age at menarche, menopause status, nulliparous status, ever use of contraceptive pill.

Abbreviations: MALT-Mucosa associated lymphoid tissues

## Potential mechanisms

This is an observational study, with no inference about causal relationship between psoriasis and appendectomy. However, some hypotheses about mechanistic mechanisms can be formulated.

Appendix has a fundamental role in immunological homeostasis as inferred from its evolutionary stability and the relative rare condition of absence of appendix in humans [15, 16]. Its immunological function is likely to be particularly important during childhood, when the appendix lymphatic follicles proliferation and activity reach their peaks [33, 34].

Appendectomy could curb the induction and proliferation of gut T regulatory cells, causing a reduction in the suppression of self-reactive T cells in the skin involved in autoimmunity as observed in murine models [35, 36]. A role of peripheral tolerance induced by the gut immunological lymphoid tissue has already been suggested in allergic skin diseases, although the exact mechanism is not fully understood [37–39]. As observed for the spleen in animal models, lymphoid in the appendix tissue could affect the skin autoimmunity through reduction of skin tolerance [40]. The appendix indirectly influences the gut immunological response, acting as a « safe house » for the regeneration of the gut microbiome during infection-induced dysbiosis [41], and via the production of IgA involved in assembly of intestinal biofilm [16, 42]. Appendectomy could disrupt these mechanisms, causing proliferation of bacterial phyla related to a gut inflammatory response [43], leading to bacterial translocation in the systemic circulation, and triggering skin immunity through the "gut-skin axis" [19, 44]. Some studies found alterations in the gut microbiome in patients with psoriasis and psoriatic arthritis, compared to healthy controls, specifically an increase in the microbiome beta-diversity [45–47].

## Limitations and strengths

The strengths of our study include its prospective design, the large size of the study population and the long follow-up time (28 years).

Ascertainment of incident cases of psoriasis was based only on the follow-up questionnaires and a full validation of our definition was not possible. There was no evidence that the probability of answering questionnaires including psoriasis-specific questions could be different among cases and non-cases. The observed incidence rate of psoriasis in our study appears to be broadly in line with estimates in previous studies in European population that vary widely typically between 40 and 300 per 100 000 persons per year, depending on the year and country [48]. Also, data from the MGEN drug reimbursement database, showed that reimbursements of drugs used only to treat psoriasis (e.g., derivatives of vitamin D) were recorded for half of our cases and reimbursements of drugs used also to treat psoriasis (i.e. such as cytokine receptor modulators, immunosuppressants, psoralens, and topical steroids) were recorded for 90% of our cases, although we did not have means to establish whether such drugs were prescribed specifically for psoriasis. Previous analyses in similar longitudinal cohorts have been based on the ascertainment of cases through self-report [3, 4], which has been shown to be reliable for case identification in a Danish study on 2333 psoriasis patients [49]. Some degree of misclassification of self-reported cases and non-cases may be present in our study but it is likely to be non-differential- i.e.- not dependent on the MALTectomy status - and therefore it would bias estimates of association toward the null.

Multivariate models allowed us to adjust for potential confounding from known factors although some residual confounding is possible. However, the results from univariate and multivariate models appear to be very similar suggesting that the association is not confounded by the covariates considered.

Biases that may arise for example from the exclusion of cases with unknown age of diagnosis and prevalent cases is likely to be marginal as suggested by the results of the sensitivity analyses based on logistic regression Also, in the sensitivity analyses we did not use the LOCF method for any of the covariates and we obtained results similar to those from the main analyses. These results provide evidence that the use of LOCF for the time-dependent covariates has no significant impact on the results.

The main limitation of the analyses on moderate-to-severe psoriasis is the modest statistical power due to the limited number of moderate-to-severe cases and the lack of information about the disease clinical manifestations.

Our definition of moderate-to-severe cases based on data from the MGEN drug reimbursements database is based on the observation that international guidelines recommend the use of systemic drugs only for moderate to moderate-to-severe psoriasis [50]. The availability of drug reimbursement data only from 2004 leaves uncertainties about the true first date of appearance of moderate-to-severe symptoms.

Data on tonsillectomy and adenoidectomy were not available after baseline and it was not possible to model them as time varying variables. Nevertheless, sensitivity analyses including appendectomy as time-varying variable did not appear to lead to significant changes in the magnitude or direction of the associations observed when the variable was considered as fixed. Furthermore, our study was not able to evaluate associations with risk of psoriasis at younger ages.

Finally, being an observational study, it is possible to establish associations but not causal links between MALTectomy and psoriasis.

## Conclusions

Our findings suggest that MALT may play a role in the development of psoriasis but such role is likely to be complex and dependent on the specific type of MALT.

These findings should be validated in other settings, especially in men and in younger populations. Should these results be confirmed, a promising new avenue in understanding the role

of MALT in the pathophysiology of cutaneous autoimmunity could be opened by using early signs of microbiome alteration before the onset of symptoms of psoriasis as biomarkers for the adoption of prophylactic measures such as microbiome transplantation or the administration of probiotics to interrupt or delay the pathogenic process.

## Supporting information

**S1 File. Supplementary tables.**
(DOCX)

## Acknowledgments

We thank all participants for providing the data used in the E3N cohort study. We also thank the members of the Operational Platform of the Exposome and Heredity team of the CESP for their technical support.

## Author Contributions

**Conceptualization:** Marco Conte, Emilie Sbidian, Gianluca Severi.

**Data curation:** Marco Conte.

**Methodology:** Marco Conte, Gianluca Severi.

**Supervision:** Gianluca Severi.

**Writing – original draft:** Marco Conte, Gianluca Severi.

**Writing – review & editing:** Marco Conte, Agnes Fournier, Joseph A. Rothwell, Marie-Christine Boutron-Ruault, Laura Baglietto, Marco Fornili, Emilie Sbidian, Gianluca Severi.

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
