## [Decision Letter · Decision Letter 0]

29 May 2023

PONE-D-23-11020MALTectomy and psoriasis risk in women: a prospective study in the French E3N prospective cohort.PLOS ONE

Dear Dr. Severi,

Thank you for submitting your manuscript to PLOS ONE. After careful consideration, we feel that it has merit but does not fully meet PLOS ONE’s publication criteria as it currently stands. Therefore, we invite you to submit a revised version of the manuscript that addresses the points raised during the review process.

We look forward to receiving your revised manuscript.

Kind regards,

Bandana Kumari, M.D, DGO, FCGP

Academic Editor

PLOS ONE

Journal Requirements:

Reviewers' comments:

Reviewer's Responses to Questions

**Comments to the Author**

1. Is the manuscript technically sound, and do the data support the conclusions?

Reviewer #1: Partly

Reviewer #2: Yes

2. Has the statistical analysis been performed appropriately and rigorously? 

Reviewer #1: Yes

Reviewer #2: Yes

3. Have the authors made all data underlying the findings in their manuscript fully available?

Reviewer #1: Yes

Reviewer #2: No

4. Is the manuscript presented in an intelligible fashion and written in standard English?

Reviewer #1: Yes

Reviewer #2: Yes

5. Review Comments to the Author

Reviewer #1: This study aimed to investigate the relationship between removal of Mucosa associated lymphoid tissue (MALTectomy) and the risk of developing psoriasis. The researchers conducted a prospective observational study using a French cohort of women. They identified incident cases of psoriasis and assessed the association with appendectomy, tonsillectomy, and adenoidectomy. The results showed that appendectomy was associated with an increased risk of psoriasis, both in overall and moderate-to-severe cases. However, no association was found with tonsillectomy or adenoidectomy. These findings suggest the need for further research to better understand the underlying mechanisms and etiology of psoriasis.

Abstract:

1. The abstract does not mention how psoriasis cases were diagnosed or confirmed. Providing information on the diagnostic criteria used or any validation steps taken would increase the reliability and validity of the findings.

2. The abstract lacks information on the characteristics of the study population, such as age range, demographic composition, and any specific inclusion or exclusion criteria. Including these details would help readers better understand the generalizability of the findings.

3. The abstract could benefit from summarizing the main statistical measures used to quantify the associations (e.g., hazard ratios with confidence intervals). Including these measures would provide a more precise understanding of the strength and precision of the observed associations.

Introduction:

1. The introduction provides a general overview of psoriasis and its risk factors but does not explicitly state the specific research gap that the study aims to address. Clearly identifying the research question or hypothesis would improve the clarity and focus of the introduction.

2. Consider providing a brief rationale for investigating the association between appendectomy and psoriasis, highlighting the existing knowledge gap or conflicting findings in previous research.

Materials and Methods:

1. The provided text lacks an introductory statement or explanation of the study design, which makes it challenging to understand the overall approach. To enhance this section:

2. Start with an introductory statement that outlines the study design and objectives.

3. Provide more information regarding the selection criteria for participants and the rationale for including them in the study.

4. Include details on the sample size calculation and justify the chosen sample size.

5. The inclusion of a visual time roadmap in the manuscript would greatly enhance the understanding of the study's timeline and provide important information regarding the duration of follow-up and the enrollment period of participants in the cohort. By including this visual representation, readers can better grasp the study's timeline and potential limitations related to the length of follow-up.

6. Describe the data collection procedures in more detail, including the methods used to collect information on covariates and outcomes.

7. Specify the statistical analysis methods used and any adjustments made for potential confounders.

8. If applicable, discuss any ethical considerations or approvals obtained for the study.

9. Describe the data collection procedures in more detail, including the methods used to collect information on covariates and outcomes. Specify the data sources, instruments used, and any validation steps taken.

10. Specify the statistical analysis methods used and any adjustments made for potential confounders. Clearly state the primary analysis approach, such as Cox proportional hazards regression models, and describe the covariates included in the analysis.

Results:

1. Include a Results section to present the study findings. Start by summarizing the key demographic characteristics of the study population.

2. Summarize the key results related to the association between appendectomy and psoriasis risk. Present the measures of association, such as hazard ratios, odds ratios, or incidence rate ratios, along with their corresponding confidence intervals and p-values.

3. Provide specific numerical results or effect sizes to quantify the observed associations. Report any statistically significant findings.

4. If applicable, the quality figures to present the data in a clear and organized manner. Ensure that the figures are properly formatted and labeled.

Discussion:

1. The discussion section is not included in the provided text, making it unclear how the study findings were interpreted and how they contribute to the existing knowledge on the topic. To enhance this section:

2. Include a Discussion section to interpret the results and discuss their implications. Start by restating the research objective or hypothesis.

3. Provide a comprehensive analysis and interpretation of the findings, considering the study's limitations and potential biases. Discuss the strengths and weaknesses of the study design and the potential impact on the results.

4. Discuss possible explanations for the observed association between appendectomy and psoriasis, such as biological or immunological pathways. Reference relevant literature and propose mechanisms that could underlie the observed relationship.

5. The manuscript highlights the association between appendectomy and an increased risk of psoriasis; however, it does not explore the underlying mechanisms or potential explanations for this association. Adding a discussion of possible biological or immunological pathways could strengthen the impact and significance of the findings. Additionally, it would be beneficial to include a high-quality figure illustrating the potential mechanisms linking appendectomy and the increased risk of psoriasis, as it would provide visual clarity and enhance the readers' understanding of the topic.

6. Compare the study findings with previous research, highlighting any conflicting or supporting evidence. Discuss the similarities and differences and provide possible explanations for discrepancies.

7. While the manuscript mentions the need for further investigation, it does not provide specific suggestions for future research directions or potential implications of the findings. Expanding on the implications and potential clinical or public health relevance would enhance the overall impact of the study.

Reviewer #2: The authors have used the NHANES data to observe an association between MALTectomy and psoriasis. In general, the article is well-written and the results support their conclusions.

The article can be accepted in its current form.

6. PLOS authors have the option to publish the peer review history of their article (what does this mean?). If published, this will include your full peer review and any attached files.

Reviewer #1: **Yes: **Manoj Khokhar

Reviewer #2: No

---

## [Author Response · Author response to Decision Letter 0]

11 Jul 2023

Editors’ comments

• We revised the entire manuscript to adhere to journal style requirements. We modified the order of Acknowledgments and Author Contributions. We modified the dimension of paragraph titles. 

• We modified the ethics statement in the methods reporting that every woman that entered the study signed a written informed consent at the study entry. We also reported that we obtained the authorization of French National Commission for Data Protection and Individual Freedom (327346-V14) and the French Advisory Committee on Information Processing in Material Research in the Field of Health (13.794) before the start of the study. Our study did not include minors.

• We reported in this letter the reasons of the restriction of data availability for the E3N cohort. Data underlying this article are made available under managed access owing to governance constraints and need to protect the privacy of study participants. Data on E3N cohort participants are available to bona fide researchers for all types of health-related research, which is in the public interest. Raw data requests should be submitted through the E3N website (www. e3n.fr) or sent to contact@e3n.fr and will be reviewed by the E3N Access Committee. Further information is provided at https://www.e3n.fr/node/78.

• Data were originally not showed because we thought they were secondary results that did not add further details to our main results. We have now included results from these sensitivity analyses in the supporting information tables S7, S8, S9, S10 and S11.

• We moved the ethics statement in the method section as instructed.

Reviewer #1 comments

• This study aimed to investigate the relationship between removal of Mucosa associated lymphoid tissue (MALTectomy) and the risk of developing psoriasis. The researchers conducted a prospective observational study using a French cohort of women. They identified incident cases of psoriasis and assessed the association with appendectomy, tonsillectomy, and adenoidectomy. The results showed that appendectomy was associated with an increased risk of psoriasis, both in overall and moderate-to-severe cases. However, no association was found with tonsillectomy or adenoidectomy. These findings suggest the need for further research to better understand the underlying mechanisms and etiology of psoriasis.

ABSTRACT

• 1. The abstract does not mention how psoriasis cases were diagnosed or confirmed. Providing information on the diagnostic criteria used or any validation steps taken would increase the reliability and validity of the findings.

• We have entirely rewritten the abstract to take into account the very useful suggestions provided by the reviewer. We reported that information was retrieved from a self-administered questionnaire and that moderate-to-severe status was established using the MGEN drug reimbursement database.

• 2. The abstract lacks information on the characteristics of the study population, such as age range, demographic composition, and any specific inclusion or exclusion criteria. Including these details would help readers better understand the generalizability of the findings. 

• We added demographic characteristics of the study into the abstract by reporting that women were born between 1925 and 1950. Under French law we are not allowed to collect information about ethnicity of our study participants but we can consider they are virtually all Caucasians. We specified exclusion and inclusion criteria as requested.

• 3. The abstract could benefit from summarizing the main statistical measures used to quantify the associations (e.g., hazard ratios with confidence intervals). Including these measures would provide a more precise understanding of the strength and precision of the observed associations.

• We rephrased the abstract to cite Cox proportional hazards models used as statistical methods for our principal analyses. We decided to not report results from logistic regression in the abstract because it was only used in sensitivity analyses. 

INTRODUCTION

• 1. The introduction provides a general overview of psoriasis and its risk factors but does not explicitly state the specific research gap that the study aims to address. Clearly identifying the research question or hypothesis would improve the clarity and focus of the introduction.

• We added a clear statement of the research gap in lines 79-80. The research gap of the study is that, despite evidences on the role of MALT in systemic immune response, we don’t have any information on their role in autoimmunity.

• 2. Consider providing a brief rationale for investigating the association between appendectomy and psoriasis, highlighting the existing knowledge gap or conflicting findings in previous research.

• We rephrased the introduction by adding a brief rationale of the study in lines 81-83. We have explained that, while laboratory studies have greatly ameliorated treatment of autoimmunity manifestations, epidemiological studies are required to understand the psoriatic pathogenesis for which we don’t have many information in the scientific literature.

MATERIALS AND METHODS

• 1. The provided text lacks an introductory statement or explanation of the study design, which makes it challenging to understand the overall approach. To enhance this section:

• We thank the reviewer and to clarify this point we added an introductory statement in the materials and methods section to explain our approach in lines 90-93. 

• 2. Start with an introductory statement that outlines the study design and objectives.

• We answered this request in the previous point

• 3. Provide more information regarding the selection criteria for participants and the rationale for including them in the study.

• There were two main selection criteria for this study: the availability of data on exposure of interest and the possibility to calculate a follow-up time to the outcome for each participant. For the first criterion we excluded women without information on MALTectomy exposures from baseline questionnaire. For the second criterion we excluded participants based on the possibility to establish the end of follow-up and then excluding all women whose date of death was not available; we also excluded all women whose date of psoriasis was not provided or was prevalent at the moment of the start of follow-up (before 1990 in overall psoriasis models and before 2004 for moderate-to-severe psoriasis models). We modified some parts of the “study population” paragraph to clarify the selection criteria. 

• 4. Include details on the sample size calculation and justify the chosen sample size.

• We did not calculate the sample size of this study because we used an existing prospective cohort for which the sample size (and the number of incident cases) was fixed. However, we performed an a posteriori statistical power calculation suggesting that our study has ample statistical power to detect even weak associations. More specifically, with the observed proportion of cases in the cohort (at least 2.5%) and a prevalence of exposed of at least 30% (the minimum prevalence across the three types of MALT), to detect a minimum hazard ratio of 1.2 (weak association) with a statistical power of 80% a cohort of at least 38 000 people would be necessary which is less than half the size of our cohort.

• 5. The inclusion of a visual time roadmap in the manuscript would greatly enhance the understanding of the study's timeline and provide important information regarding the duration of follow-up and the enrollment period of participants in the cohort. By including this visual representation, readers can better grasp the study's timeline and potential limitations related to the length of follow-up.

• We have drawn a timeline that include the duration of the follow-up for both overall and moderate-to-severe models. In this figure we reported year below and the order of questionnaires output above. We think this illustration would permit to appreciate the length of the study follow-up. 

• 6. Describe the data collection procedures in more detail, including the methods used to collect information on covariates and outcomes.

• We reported details of outcomes and covariates in the paragraphs “Assessment of MALTectomy” (lines 127-133) and “Assessment of covariates” respectively (lines 135-147).

• 7. Specify the statistical analysis methods used and any adjustments made for potential confounders. 

• We described the Cox proportional hazards model used in the study in the “statistical analyses” paragraph. We rearranged this paragraph to make its content clearer.

• 8. If applicable, discuss any ethical considerations or approvals obtained for the study.

• Ethical considerations were present in a statement that was included in the acknowledgements and that we have now moved into the methods section at lines 224-229.

• 9. Describe the data collection procedures in more detail, including the methods used to collect information on covariates and outcomes. Specify the data sources, instruments used, and any validation steps taken. 

• These data were reported in “The E3N cohort” paragraph where we detailed that information was collected through self-administered questionnaires sent by mail every two or three years. Questionnaires were then scanned and reviewed by E3N data managers. Many validation studies on the reliability of extracted information have been conducted as detailed in a previous article (Clavel-Chapelon F; E3N Study Group. Cohort Profile: The French E3N Cohort Study. Int J Epidemiol. 2015 Jun;44(3):801-9. doi: 10.1093/ije/dyu184. Epub 2014 Sep 10. PMID: 25212479).

• 10. Specify the statistical analysis methods used and any adjustments made for potential confounders. Clearly state the primary analysis approach, such as Cox proportional hazards regression models, and describe the covariates included in the analysis.

• We reported this information in the “statistical analysis” paragraph where we defined the start and end of follow up time for Cox proportional hazard regression models used in our main analyses. We also reported that we used Schoenfeld residuals to test the proportional hazards assumption. Adjustment’s variables were initially described in “assessment of covariates” paragraph (lines 135-147) and then further explained in the “Statistical analysis” paragraph (lines 191-204).

RESULTS

• 1. Include a Results section to present the study findings. Start by summarizing the key demographic characteristics of the study population. 

• The first paragraph of results includes a description of the key characteristics of study population. We added a sentence and data in table 1 reporting the mean age at baseline, the mean time between baseline and psoriasis diagnosis and mean age at psoriasis diagnosis.

• 2. Summarize the key results related to the association between appendectomy and psoriasis risk. Present the measures of association, such as hazard ratios, odds ratios, or incidence rate ratios, along with their corresponding confidence intervals and p-values.

• We reported results of associations between MALTectomy and psoriasis risk between lines 259-313 where we also reported Hazard ratios and Odds ratio. We did not report p-values in the text, but they are reported in tables 1-4.

• 3. Provide specific numerical results or effect sizes to quantify the observed associations. Report any statistically significant findings. 

• We reported statistically significant results obtained from our study in lines 259-296. The only significant result was the association between appendectomy and psoriasis risk. This association was observed in overall model while moderate-to-severe model showed a suggestive association even if intervals contained 1. We reported data from logistic regressions performed as sensitive analysis that showed that the association between appendectomy and psoriasis risk is observed also when we introduced prevalent cases and cases without a diagnostic date. Results from all performed analyses are reported in tables 1-4 or in supporting information tables S2-S11.

• 4. If applicable, the quality figures to present the data in a clear and organized manner. Ensure that the figures are properly formatted and labeled. 

• We revised all included images of the study to harmonize format and contents.

DISCUSSION

• 1. The discussion section is not included in the provided text, making it unclear how the study findings were interpreted and how they contribute to the existing knowledge on the topic. To enhance this section: 

• The discussion was present in the manuscript and included different sections to help readability. We have made some modifications and additions to clarify some key points. (See responses to the points below). 

• 2. Include

---

## [Decision Letter · Decision Letter 1]

29 Jan 2024

PONE-D-23-11020R1MALTectomy and psoriasis risk in women: a prospective study in the French E3N prospective cohort.PLOS ONE

Dear Dr. Severi,

Thank you for submitting your manuscript to PLOS ONE. After careful consideration, we feel that it has merit but does not fully meet PLOS ONE’s publication criteria as it currently stands. Therefore, we invite you to submit a revised version of the manuscript that addresses the points raised during the review process.

We look forward to receiving your revised manuscript.

Kind regards,

Karthika Paul

Academic Editor

PLOS ONE

Reviewers' comments:

Reviewer's Responses to Questions

**Comments to the Author**

1. If the authors have adequately addressed your comments raised in a previous round of review and you feel that this manuscript is now acceptable for publication, you may indicate that here to bypass the “Comments to the Author” section, enter your conflict of interest statement in the “Confidential to Editor” section, and submit your "Accept" recommendation.

Reviewer #1: All comments have been addressed

2. Is the manuscript technically sound, and do the data support the conclusions?

Reviewer #1: Yes

3. Has the statistical analysis been performed appropriately and rigorously? 

Reviewer #1: Yes

4. Have the authors made all data underlying the findings in their manuscript fully available?

Reviewer #1: Yes

5. Is the manuscript presented in an intelligible fashion and written in standard English?

Reviewer #1: Yes

6. Review Comments to the Author

Reviewer #1: The article provides intriguing findings regarding the association between MALTectomy and psoriasis risk. However, it would benefit from a more detailed explanation of statistical methods, a thorough discussion of potential biases, and a cautious interpretation of the results in the context of causation. Additionally, the limitations and generalizability of the study should be discussed more comprehensively, and recommendations for future research should be provided.

1. The article addresses an intriguing research gap by investigating the role of MALT tissues in the development of autoimmune responses related to psoriasis. This study is significant because it explores a relatively uncharted area, potentially shedding light on psoriasis etiology.

2. While the use of the E3N cohort is appropriate for this type of research, the article should include a discussion of potential limitations associated with self-reported data and data from the MGEN drug reimbursement database. This acknowledgment would enhance transparency.

3. The article explains the exclusion criteria but should discuss the implications of excluding specific participants in detail. A more comprehensive discussion of potential biases introduced by these exclusions is warranted.

4. The article includes a range of covariates and potential risk factors in the analysis. However, offering a more detailed explanation of why specific variables were chosen and their potential impact on the outcomes would enhance the research design's clarity.

5. The use of the "last observation carried forward" method for handling missing data is mentioned briefly. A more in-depth discussion of the implications of this method, especially in the sensitivity analysis, is warranted.

6. While the statistical methods are described, more detail on model selection and the criteria for including or excluding specific variables would enhance the transparency of the analysis.

7. The sensitivity analysis is a strong point of the study. However, explicitly stating the rationale for including cases with an unknown age of diagnosis and prevalent cases enhances the interpretation of these results.

8. The article should address potential sources of bias, such as selection bias, recall bias, and reporting bias, which can be inherent in observational cohort studies using self-reported data. This discussion is essential to provide a comprehensive view of the study's potential limitations.

9. It's crucial to emphasize that the study establishes associations rather than causation. This distinction should be made explicit throughout the article to ensure a cautious interpretation of the findings.

10. Providing specific recommendations for future research and discussing the potential implications for clinical practice and public health would enhance the study's impact and guide further investigations.

7. PLOS authors have the option to publish the peer review history of their article (what does this mean?). If published, this will include your full peer review and any attached files.

Reviewer #1: **Yes: **Dr. Manoj Khokhar

---

## [Author Response · Author response to Decision Letter 1]

10 Mar 2024

Reviewer #1 comments

The article provides intriguing findings regarding the association between MALTectomy and psoriasis risk. However, it would benefit from a more detailed explanation of statistical methods, a thorough discussion of potential biases, and a cautious interpretation of the results in the context of causation. Additionally, the limitations and generalizability of the study should be discussed more comprehensively, and recommendations for future research should be provided.

The article addresses an intriguing research gap by investigating the role of MALT tissues in the development of autoimmune responses related to psoriasis. This study is significant because it explores a relatively uncharted area, potentially shedding light on psoriasis etiology.

• 1. While the use of the E3N cohort is appropriate for this type of research, the article should include a discussion of potential limitations associated with self-reported data and data from the MGEN drug reimbursement database. This acknowledgment would enhance transparency.

• A section on the study limitations was included in the article but we have extended it to include the following considerations. The use we made of self-reported psoriasis cases may have led to some misclassifications of cases and non-cases that is likely to be relatively limited due to the following reasons : i) the incidence of psoriasis in our population is similar of that reported in other European studies with the same demographics; ii) for most reported cases we found claims for the reimbursements of drugs used in psoriasis in the MGEN drugs reimbursement database; and iii) self-reported information was used in observational studies in other cohorts and some of these studies showed a good validity of self-reported diagnosis when compared with a dermatologist visit performed in a subset of cases. It is still possible that a certain degree of misclassification of self-reported cases is present, but such misclassification would bias estimates of association toward the null, being it unlikely to be differential relative to the MALTectomy status. A limitation of data on drug reimbursement is that no information is available in the MGEN dataset before 2004 that leaves uncertainties about the start of the treatment of psoriasis. Also, except for vitamin A and D topical agents that are specific for the treatment of psoriasis other drugs may be used also for other purposes. Despite this limitation, the observation that 90% of self-reported cases presented prescriptions for drugs used to treat psoriasis provides support for the use of self-reported cases. 

• 2. The article explains the exclusion criteria but should discuss the implications of excluding specific participants in detail. A more comprehensive discussion of potential biases introduced by these exclusions is warranted.

• We have expanded the discussion of the potential biases, specifically addressing the potential consequences of excluding participants because they were lost to follow-up or for other reasons we detailed in the text and the flow charts. Women lost to follow-up had a similar profile of MALTectomy compared with women included in the study making it unlikely an impact of loss to follow-up on our results. Prevalent cases and cases without a date of diagnosis were excluded from the main analysis but we included them in sensitivity analyses based on logistic regression that generated results similar to the main analyses and provided support to exclude that such exclusions induced significant biases.

• 3. The article includes a range of covariates and potential risk factors in the analysis. However, offering a more detailed explanation of why specific variables were chosen and their potential impact on the outcomes would enhance the research design's clarity.

• The choice of variables included in the models is based on the availability of data in the E3N cohort about any factor that may be associated to both MALTectomy and psoriasis. We included such factors to try to reduce potential confounding and thus better isolate the effect of MALtectomy on psoriasis risk. The covariates were chosen based on knowledge from the literature and based on hypotheses about potential effects of sociodemographic, hormonal and reproductive factors on the studied association. 

For example, BMI, smoking, hypertension, diabetes, and depression were included because there was evidence in the literature of an association between these variables and psoriasis. Other variables, such as marital status and level of education, were included on the hypothesis that sociodemographic factors may confound the association between MALTectomy and psoriasis while we included hormonal and reproductive factors such as age of menarche, parity, and ever use of the contraceptive pill that may be potential confounding factors in the onset of psoriasis as reported in the literature. 

Given that the HR estimates in the univariate models were nearly identical to those obtained from the multivariable models we can conclude that the adjustment for the covariables in our model did not affect the association between MALTectomy and risk of psoriasis. 

We rephrased the assessment of covariates paragraph to better explain this point.

• 4. The use of the "last observation carried forward" method for handling missing data is mentioned briefly. A more in-depth discussion of the implications of this method, especially in the sensitivity analysis, is warranted.

• The LOCF method was used only for the time-dependent covariables included in the multivariable models while it was not used for the variables of interest and outcome. In the sensitivity analyses based on logistic regression we did not use the LOCF method for any of the covariates and we obtained results similar to those from the main analyses. These results provide evidence that the use of LOCF for the time-dependent covariates has no significant impact on the results. 

• 5. While the statistical methods are described, more detail on model selection and the criteria for including or excluding specific variables would enhance the transparency of the analysis.

• The choice of Cox semiparametric model was made because it makes it possible to quantify the relationship between MALTectomy and psoriasis by considering the time to event. We discussed earlier the reasons we included the different covariates in the multivariable models (see point 3).

• 6. The sensitivity analysis is a strong point of the study. However, explicitly stating the rationale for including cases with an unknown age of diagnosis and prevalent cases enhances the interpretation of these results.

• As explained earlier, the aim of the sensitivity analyses to evaluate the influence of prevalent cases and cases without date of diagnosis on the associations tested in the main models from which we excluded such cases. Our main objective was to test the association with incident psoriasis (cases diagnosed during follow-up starting at the establishment of the cohort in 1990) and for this reason prevalent cases were excluded from the main analysis that were based on Cox models. Cases without a date of diagnosis were excluded because we could not assess whether they were prevalent or incident. Also, prevalent cases, diagnosed before the inclusion in the cohort and therefore several years or decades earlier, are likely to be reported with greater uncertainty than incident cases. The observation that the sensitivity analyses based on logistic regression and including also prevalent cases and cases without a date of diagnosis provided results similar to those from the Cox models suggest that results are robust with respect to the choice we have made for the main analyses. We have improved the paragraph explaining this point in the materials and methods.

• 7. The article should address potential sources of bias, such as selection bias, recall bias, and reporting bias, which can be inherent in observational cohort studies using self-reported data. This discussion is essential to provide a comprehensive view of the study's potential limitations.

• The different types of bias that may be at play in our study have been addressed and discussed earlier and through modifications in the text of the manuscript (see in particular responses to point 1,2 and 6).

• 8. It's crucial to emphasize that the study establishes associations rather than causation. This distinction should be made explicit throughout the article to ensure a cautious interpretation of the findings.

• We have made modifications to caution about the results of an observational study stressing in the discussion the distinction between the associations we observed and causal relationships that cannot be established in this type of study.

• 9. Providing specific recommendations for future research and discussing the potential implications for clinical practice and public health would enhance the study's impact and guide further investigations.

• We modified the conclusion of the article to add recommendations for future research. We note in the conclusions that our results suggest that an interesting target for research on psoriasis may be the dysbiosis process induced by appendectomy. Early signs of microbiome alteration before the onset of disease symptoms should be explored as biomarkers for the adoption of prophylactic measures such as microbiome transplantation or the administration of probiotics to interrupt or delay the pathogenic process.

---

## [Decision Letter · Decision Letter 2]

9 Sep 2024

MALTectomy and psoriasis risk in women: a prospective study in the French E3N prospective cohort.

PONE-D-23-11020R2

Dear Dr. Severi,

We’re pleased to inform you that your manuscript has been judged scientifically suitable for publication and will be formally accepted for publication once it meets all outstanding technical requirements.

Kind regards,

Karthika Paul

Academic Editor

PLOS ONE

Additional Editor Comments (optional):

Accepted for the publication

Reviewers' comments:

Reviewer's Responses to Questions

**Comments to the Author**

1. If the authors have adequately addressed your comments raised in a previous round of review and you feel that this manuscript is now acceptable for publication, you may indicate that here to bypass the “Comments to the Author” section, enter your conflict of interest statement in the “Confidential to Editor” section, and submit your "Accept" recommendation.

Reviewer #1: All comments have been addressed

Reviewer #3: (No Response)

2. Is the manuscript technically sound, and do the data support the conclusions?

Reviewer #1: Yes

Reviewer #3: Yes

3. Has the statistical analysis been performed appropriately and rigorously? 

Reviewer #1: Yes

Reviewer #3: Yes

4. Have the authors made all data underlying the findings in their manuscript fully available?

Reviewer #1: Yes

Reviewer #3: Yes

5. Is the manuscript presented in an intelligible fashion and written in standard English?

Reviewer #1: Yes

Reviewer #3: Yes

6. Review Comments to the Author

Reviewer #1: (No Response)

Reviewer #3: (No Response)

7. PLOS authors have the option to publish the peer review history of their article (what does this mean?). If published, this will include your full peer review and any attached files.

Reviewer #1: **Yes: **Dr. Manoj Khokhar

Reviewer #3: No

---

## [Editor Report · Acceptance letter]

19 Sep 2024

PONE-D-23-11020R2 

PLOS ONE

Dear Dr. Severi, 

I'm pleased to inform you that your manuscript has been deemed suitable for publication in PLOS ONE. Congratulations! Your manuscript is now being handed over to our production team.

Kind regards, 

on behalf of

Dr. Karthika Paul 

Academic Editor

PLOS ONE